# POD-Galerkin FSI Analysis for Flapping Motion

**DOI:** 10.3390/biomimetics8070523

**Published:** 2023-11-03

**Authors:** Shigeki Kaneko, Shinobu Yoshimura

**Affiliations:** Department of Systems Innovations, School of Engineering, The University of Tokyo, Tokyo 113-8656, Japan; yoshi@sys.t.u-tokyo.ac.jp

**Keywords:** reduced-order model, proper orthogonal decomposition, fluid–structure interaction, partitioned iterative coupled analysis, flapping motion

## Abstract

FSI simulations of flapping motions have been widely investigated to develop a flapping-wing micro air vehicle. Because an intensive parametric study is important for the product design, a computationally efficient model is required. The purpose of the present study was to develop a reduced-order model of flapping motion. Among the various methods available to solve FSI problems, we employed the Dirichlet–Neumann partitioned iterative method, in which three sub-systems (fluid mesh update, fluid analysis, and structural analysis) are executed. In the proposed analysis system, first, snapshot data of structural displacement, fluid velocity, fluid pressure, and displacement for the fluid mesh update were collected from a high-fidelity FSI analysis. Then, the snapshot data were used to create low-dimensional surrogate systems of the above three sub-systems based on the POD under Galerkin projection (i.e., the POD-Galerkin method). In numerical examples, we considered a two-dimensional FSI problem of simplified flapping motion. The problem was described via two parameters: frequency and amplitude of flapping motion. We demonstrated the effectiveness of the presented reduced-order model in significantly reducing computational time while preserving the desired accuracy.

## 1. Introduction

Birds and insects achieve amazing aerodynamic performance using flapping motions, which are known to be efficient in the low Reynolds number regime. Inspired by flapping flight observed in nature, FWMAVs have been actively investigated over the last two decades. Although FWMAVs have not yet been made practical, some prototypes, such as Delfly [1] and RoboBee [2], have been developed.

To accelerate the design of devices and mechanisms, numerical simulations are effective. Many researchers have worked on numerical studies related to FWMAVs [3,4]. In our previous studies, we performed FEM-based FSI analysis to simulate 3D hovering flight with flexible flapping wings [5,6]. Then, we investigated the feasibility of FWMAVs in the Earth and Martian environments.

Simulations of FWMAVs usually need a lot of computational resources because flapping motions are complex phenomena, where flapping wings and the surrounding fluid interfere mutually. Although recent improvements in computer performance have enabled us to perform detailed simulations using many DOFs within a practical computational time, a computationally efficient FWMAV model is required because an intensive parametric study for a wide range of combinations of various flapping motions, such as flapping, pitching, and lead-lag, is essential for the realization of FWMAVs.

To reduce the computational burden, one promising approach is ROM. For problems with extremely high-dimensional parametrized systems, lower-dimensional manifolds with representative key features are sought. ROM comprises an offline phase and an online phase. In the offline phase, pre-computable and computationally expensive procedures are performed. Sufficient training datasets (i.e., snapshot data) are collected from high-fidelity analysis, and data compression techniques, such as POD [7], are applied to obtain reduced-order bases. In the online phase, a computationally efficient reduced-order model is constructed using reduced-order bases, and the model is used to accelerate predictive simulations. The POD method has been widely applied in conjunction with Galerkin projection to build reduced-order models for a wide range of engineering problems, such as thermal–visco-plastic deformation behavior [8], unsteady turbulent incompressible flow [9], contact problems [10], and crack propagation [11], among others.

The methods to solve FSI problems are classified into two types: interface capturing [12,13] and interface tracking [14,15]. Interface-capturing methods have a great advantage in handling topology change. In contrast, interface-tracking methods are accurate due to the precise representation of fluid–structure interfaces. Interface-tracking methods are implemented based on monolithic [16] or partitioned approaches [17]. Monolithic approaches are generally known to be accurate, while partitioned approaches are easy to implement because they allow existing solvers to be used. Iterations can be introduced into partitioned approaches to achieve the fully implicit treatment of the coupling conditions [18]. The accuracy of the partitioned iterative method is comparable to that of the monolithic approaches. Based on the Dirichlet–Neumann approach [19], our research group has been developing a partitioned iterative FSI analysis system [5,18,20].

Regarding existing studies on the applications of the POD-Galerkin method to FSI analysis, many of them have followed interface-capturing approaches, such as the immersed boundary method and the fictitious domain method [21,22,23,24,25]. In contrast, few studies have addressed ROM for interface-tracking FSI analysis [26,27]. Ballarin and Rozza proposed ROM for monolithic FSI analysis based on the ALE method, which is one of the interface-tracking approaches [26]. Nonino et al. proposed ROM for partitioned FSI analysis based on the ALE method [27]. They employed a Chorin–Temam projection scheme for the Navier–Stokes equations [28] and the semi-implicit treatment of the coupling conditions.

The purpose of the present study was to develop a reduced-order model of flapping-motion FSI problems. Although some researchers have developed such models [23,24], their studies were based on immersed boundary methods. Unlike these existing studies, we employed the interface-tracking method. In the present study, the POD-Galerkin method was applied to the ALE-method-based partitioned iterative FSI analysis.

An outline of this paper is as follows: The high-fidelity computational model constructed based on FEM is given in Section 2. The POD-Galerkin procedure is described in Section 3. In Section 4, numerical examples are presented to demonstrate the effectiveness of the proposed reduced-order model. Finally, the conclusions are given in Section 5.

## 2. High-Fidelity FSI Analysis for Flapping Motion

### 2.1. Problem Setting

In the present study, all quantities are in SI units.

Here, we specify an FSI problem of flapping motion. Figure 1 shows a schematic view of the 2D FSI problem, which we analyze in the present study. Due to the difficulty of visualizing the thin structure, Figure 1 does not show the target problem in true scale. An elastic structure is placed at the center of a non-flowing fluid. The upper edge of the structure (DA) is assumed to be clamped, and the forced displacement, U, is imposed on edge DA so that the structure is swung left and right. $U={\left[U_{x}\left(t\right)\;U_{y}\left(t\right)\right]}^{T}$ is defined as
(1)$$U_{x}\left(t\right)=A\sin\left(2\pi ft\right),\;U_{y}\left(t\right)=0,$$
where *A*, *f*, and *t* denote the amplitude, frequency, and time, respectively. The domains of the structure and the fluid are denoted by $\Omega_{S}$ and $\Omega_{F}$, respectively. The interface between $\Omega_{S}$ and $\Omega_{F}$ is the fluid–structure interface, denoted by $\Gamma_{\mathrm{FSI}}$, which is composed of four edges: AB, BC, CD, and DA. The boundary of the fluid domain is denoted by $\Gamma_{F}$, which is composed of four edges: EF, FG, GH, and HE. In the present study, a traction-free condition is imposed on $\Gamma_{\mathrm{FSI}}$. Note that a 2D simplified flapping problem where a thin structure is vibrated in a non-flowing fluid has been widely used for fundamental research on the aerodynamics of hovering [29,30].

### 2.2. Governing Equations

#### 2.2.1. Equations for Fluid

Under the assumption of incompressible viscous flow, the fluid dynamics are governed by the Navier–Stokes equation in an ALE frame of reference: (2)$$\rho^{F}\left({\left.\frac{\partial v^{F}}{\partial t}\right|}_{\chi }+\left(v^{F}-{\hat{v}}^{F}\right)\cdot \nabla_{x}v^{F}\right)-\nabla_{x}\cdot \sigma^{F}=\rho^{F}b^{F},$$
with the following continuity equation: (3)$$\nabla_{x}\cdot v^{F}=0,$$
where $\rho^{F}$ is the fluid density, $v^{F}$ is the fluid velocity vector, ${\hat{v}}^{F}$ is the mesh velocity vector, $\sigma^{F}$ is the fluid Cauchy stress tensor, and $b^{F}$ is the body force vector applied to the fluid. In the present study, $b^{F}=0$. The nabla operator, $\nabla_{x}$, refers to the current configuration. $\frac{\partial v^{F}}{\partial t}{|}_{\chi }$ represents the referential time derivative of the solution in the spatial configuration. A Newtonian fluid is assumed, and $\sigma^{F}$ is defined as follows: (4)$$\sigma^{F}=-p^{F}I+\mu \left(\nabla_{x}v^{F}+{\left(\nabla_{x}v^{F}\right)}^{T}\right),$$
where $p^{F}$ is the fluid pressure, I is the unit tensor, and μ is the fluid viscosity.

The stress field, $\sigma^{F}$, is subjected to the following traction-free condition: (5)$$\sigma^{F}\cdot n^{F}=0\;\mathrm{on}\;\Gamma_{F},$$
where $n^{F}$ is the outward normal vector.

#### 2.2.2. Equations for Structure

The mechanical behavior is governed by the following Cauchy momentum equation: (6)$$\rho_{0}^{S}\frac{D^{2}u^{S}}{Dt^{2}}-\nabla_{X}\cdot \left(S\cdot F^{T}\right)=\rho_{0}b_{0}^{S},$$
where $\rho_{0}^{S}$ is the structural density, $u^{S}$ is the displacement vector, $b_{0}^{S}$ is the body force vector, F is the deformation gradient, S is the second Piola–Kirchhoff stress, and $\frac{D}{Dt}$ is the material derivative. In the present study, $b_{0}^{S}=0$. The nabla operator, $\nabla_{X}$, refers to the initial configuration. The governing equation is described with respect to the initial configuration (denoted by subscript 0).

The constitutive equations are modeled by the following linear relations: (7)S=C:E,
where $E=\frac{1}{2}\left(\nabla_{X}u^{S}+{\left(\nabla_{X}u^{S}\right)}^{T}+{\left(\nabla_{X}u^{S}\right)}^{T}\cdot \nabla_{X}u^{S}\right)$ is the Green–Lagrange strain and C is the fourth-order elastic tensor.

The forced displacement is prescribed on edge DA as a Dirichlet boundary condition: (8)$$u^{S}=U\;\mathrm{on}\;\mathrm{DA}.$$

#### 2.2.3. Equations for Interaction Conditions on the FSI Interface

The equilibrium force and geometric compatibility at the FSI interface $\Gamma_{\mathrm{FSI}}$ are expressed as follows: (9)$$\sigma^{S}\cdot n^{S}=-\sigma^{F}\cdot n^{F}=F^{\mathrm{FSI}}\;\mathrm{on}\;\Gamma_{\mathrm{FSI}},$$
(10)$$v^{F}=\frac{Du^{S}}{Dt}\;\mathrm{on}\;\Gamma_{\mathrm{FSI}},$$
where $\sigma^{S}$ is the Cauchy stress tensor for the structure, $n^{S}$ is the outward normal vector, and $F^{\mathrm{FSI}}$ is the fluid force.

#### 2.2.4. Equations for Fluid Mesh Update

Because the ALE method is used, it is necessary to move the fluid mesh to match the motion of the structure. For the mesh update, pseudo-elastic smoothing [31] is employed, as in many other studies that have used FSI analysis [16]. We now explain how pseudo-elastic smoothing updates the fluid mesh when the fluid coordinate at time *t*, denoted by $x_{t}^{M}$, and the displacement on $\Gamma_{\mathrm{FSI}}$ at time t+Δt, denoted by $u_{t+\Delta t}^{S}$, are given. In pseudo-elastic smoothing, mesh deformation in the fluid domain is governed by the linear elastic equation: (11)$$\nabla_{X^{\prime }}\cdot \sigma^{M}=0\;\mathrm{in}\;\Omega_{F},$$
where $\sigma^{M}$ is the Cauchy stress tensor. Here, the initial configuration for this elasticity problem is $x_{t}^{M}$. To distinguish the Lagrangian description in the equations for structure (Section 2.2.2), we introduce $\nabla_{X^{\prime }}$. For linear elasticity, $\sigma^{M}$ is defined as
(12)$$\sigma^{M}=C^{M}:E^{M},$$
where $C^{M}$ is a fourth-order elasticity tensor and $E^{M}=\frac{1}{2}\left(\nabla_{X^{\prime }}u^{M}+{\left(\nabla_{X^{\prime }}u^{M}\right)}^{T}\right)$ is the infinitesimal strain tensor. For the determination of $C^{M}$ in the fluid mesh update, Jacobian-based stiffening is employed. For the details of the method, we refer readers to [5]. In the pseudo-elastic smoothing scheme, there is no Neumann boundary condition. The following Dirichlet boundary condition is prescribed: (13)$$\left\{\begin{array}{cc} u^{M}=u_{t+\Delta t}^{S}-u_{t}^{S} & \mathrm{on}\;\Gamma_{\mathrm{FSI}},\;\\ u^{M}=0 & \mathrm{on}\;\Gamma_{F}, \end{array}\right.$$
where $u^{M}$ is the displacement of fluid nodes from the configuration $x_{t}^{M}$.

$u^{M}$ is calculated by solving Equation (Equation 11) with the boundary condition (Equation (Equation 13)). $x_{t+\Delta t}^{M}$ is obtained as
(14)$$x_{t+\Delta t}^{M}=x_{t}^{M}+u^{M}.$$

The fluid mesh velocity at time t+Δt, denoted by ${\hat{v}}_{t+\Delta t}^{F}$, is obtained as
(15)$${\hat{v}}_{t+\Delta t}^{F}=\frac{\left(x_{t+\Delta t}^{M}-x_{t}^{M}\right)}{\Delta t}.$$

Note that the mesh velocity corresponds to the velocity of the structure on $\Gamma_{\mathrm{FSI}}$.

### 2.3. Partitioned Iterative FSI Analysis

To solve Equations (Equation 2)–(Equation 15), we employ the partitioned iterative method, where sub-analysis systems are executed while satisfying the interface conditions (Equations (Equation 9) and (Equation 10)). In the present study, we used three systems, namely, an analysis system for the fluid mesh update, one for fluids, and one for structures. In this subsection, we explain the detailed procedures of the partitioned iterative method based on the Dirichlet–Neumann approach [19].

For spatial discretization, we employed the FEM. $u^{M}$, $v^{F}$, $p^{F}$, and $u^{S}$ are approximated as
(16)$$u^{M}\approx u_{h}^{M}=N^{F}d^{M},\;v^{F}\approx v_{h}^{F}=N^{F}d^{V},\;p^{F}\approx p_{h}^{F}=N^{P}d^{P},\;u^{S}\approx u_{h}^{S}=N^{S}d^{S},$$
where the subscript *h* denotes the numerical approximation and $d^{M}$, $d^{V}$, $d^{P}$, and $d^{S}$ are the discretized forms of the displacement of fluid nodes, the fluid velocity, the fluid pressure, and the structural displacement, respectively. $N^{F}$, $N^{P}$, and $N^{S}$ are defined as
$$\begin{array}{cc} \; & N^{F}=\left[\begin{array}{cccccccc} N_{1}^{F} & 0 & \cdots & N_{I}^{F} & 0 & \cdots & N_{n^{F}}^{F} & 0\\ 0 & N_{1}^{F} & \cdots & 0 & N_{I}^{F} & \cdots & 0 & N_{n^{F}}^{F} \end{array}\right],\\ \; & N^{P}=\left[\begin{array}{ccccc} N_{1}^{F} & \cdots & N_{I}^{F} & \cdots & N_{n^{F}}^{F} \end{array}\right],\\ \; & N^{S}=\left[\begin{array}{cccccccc} N_{1}^{S} & 0 & \cdots & N_{I}^{S} & 0 & \cdots & N_{n^{S}}^{S} & 0\\ 0 & N_{1}^{S} & \cdots & 0 & N_{I}^{S} & \cdots & 0 & N_{n^{S}}^{S} \end{array}\right], \end{array}$$
where $n^{F}$ and $n^{S}$ are the numbers of fluid and structural nodes and $N_{I}^{F(S)}$ denotes the shape function associated with the *I*-th fluid (structural) node. $d^{M}$, $d^{V}$, $d^{P}$, and $d^{S}$ are defined as
$$\begin{array}{cc} \; & d^{M}={\left[\begin{array}{cccccccc} d_{1x}^{M} & d_{1y}^{M} & \cdots & d_{Ix}^{M} & d_{Iy}^{M} & \cdots & d_{n^{F}x}^{M} & d_{n^{F}y}^{M} \end{array}\right]}^{T},\\ \; & d^{V}={\left[\begin{array}{cccccccc} d_{1x}^{V} & d_{1y}^{V} & \cdots & d_{Ix}^{V} & d_{Iy}^{V} & \cdots & d_{n^{F}x}^{V} & d_{n^{F}y}^{V} \end{array}\right]}^{T},\\ \; & d^{P}={\left[\begin{array}{ccccc} d_{1}^{P} & \cdots & d_{I}^{P} & \cdots & d_{n^{F}}^{P} \end{array}\right]}^{T},\\ \; & d^{S}={\left[\begin{array}{cccccccc} d_{1x}^{S} & d_{1y}^{S} & \cdots & d_{Ix}^{S} & d_{Iy}^{S} & \cdots & d_{n^{S}x}^{S} & d_{n^{S}y}^{S} \end{array}\right]}^{T}, \end{array}$$
where the subscript *I* means the value is associated with the *I*-th node and the subscript x(y) indicates a x(y) component of the vector.

In the fluid mesh update, FEM is employed for spatial discretization. Then, the following weak form with the Dirichlet boundary condition (Equation (Equation 13)) is solved: find $u_{h}^{M}\in V^{M}+G^{M}$, such that $\forall w_{h}^{M}\in V^{M}$: (17)$$\int_{\Omega_{F}}\left(\frac{1}{2}\left(\nabla_{X^{\prime }}w_{h}^{M}+\nabla_{X^{\prime }}{w_{h}^{M}}^{T}\right)\right):C^{M}:\left(\frac{1}{2}\left(\nabla_{X^{\prime }}u_{h}^{M}+\nabla_{X^{\prime }}{u_{h}^{M}}^{T}\right)\right)d\Omega =0,$$
where $w^{M}$ denotes the weight function. The function space, $V^{M}$, is defined as
(18)$$V^{M}=\left\{w_{h}^{M}|\exists e^{M}={\left[e_{1x}^{M}\;e_{1y}^{M}\;\cdots \;e_{Ix}^{M}\;e_{Iy}^{M}\;\cdots \;e_{n^{F}x}^{M}\;e_{n^{F}y}^{M}\right]}^{T}\in R^{2n^{F}},w_{h}^{M}=N^{F}e^{M},e_{Ix}^{M}=e_{Iy}^{M}=0\;\mathrm{for}\;I\in \eta^{M}\right\},$$
where $\eta^{M}$ is a set of fluid nodes placed on $\Gamma_{F}$ and $\Gamma_{\mathrm{FSI}}$. $G^{M}=N^{F}g^{M}$ is a lifting function, which is needed to satisfy the Dirichlet boundary condition [27,32,33]. $g^{M}$ is defined as
(19)$$g^{M}={\left[g_{1x}^{M}\;g_{1y}^{M}\;\cdots \;g_{Ix}^{M}\;g_{Iy}^{M}\;\cdots \;g_{n^{F}x}^{M}\;g_{n^{F}y}^{M}\right]}^{T},\left\{\begin{array}{cc} {\left[g_{Ix}^{M}\;g_{Iy}^{M}\right]}^{T}={u_{h}^{S}}_{t+\Delta t}\left(\Psi \left(x_{I}^{M}\right)\right)-{u_{h}^{S}}_{t}\left(\Psi \left(x_{I}^{M}\right)\right) & I\in \eta^{\mathrm{FSI}},\\ {\left[g_{Ix}^{M}\;g_{Iy}^{M}\right]}^{T}=0 & I\notin \eta^{\mathrm{FSI}}, \end{array}\right.$$
where $x_{I}^{M}$ is the *I*-th fluid node, $\eta^{\mathrm{FSI}}$ is a set of fluid nodes placed on $\Gamma_{\mathrm{FSI}}$, and Ψ is a one-to-one mapping relating the Lagrangian frame for the fluid mesh update to the Lagrangian frame for the structural analysis. Equation (Equation 19) is discretized as follows: (20)$${e^{M}}^{T}K^{M}d^{M}=0\;\left(\forall e^{M}\in R^{2n^{F}},e_{Ix}^{M}=e_{Iy}^{M}=0\;\mathrm{for}\;I\in \eta^{M}\right),$$
where $K^{M}\in R^{2n^{F}\times 2n^{F}}$ is the stiffness matrix. After calculating the displacement of fluid nodes, the fluid mesh coordinate, $x^{M}$, and the mesh velocity, ${\hat{v}}^{F}$, are calculated based on Equations (Equation 14) and (Equation 15). The fluid mesh update procedure, M, is performed as follows: (21)$$\left({x_{h}^{M}}_{t+\Delta t},{\hat{v}}_{h}^{F}t+\Delta t\right)=M\left({u_{h}^{S}}_{t+\Delta t}\right).$$

In the flow analysis, the FEM is employed for spatial discretization. To avoid instabilities, the Petrov–Galerkin method (SUPG and PSPG methods [34]) is employed. Because of the non-slip condition (Equation (Equation 10)), the fluid velocity corresponds to the mesh velocity on $\Gamma_{\mathrm{FSI}}$: (22)$$v^{F}={\hat{v}}^{F}\;\mathrm{on}\;\Gamma_{\mathrm{FSI}},$$
which is imposed as a Dirichlet boundary condition in the flow analysis. In addition, the traction-free boundary condition (Equation (Equation 5)) is prescribed on $\Gamma_{F}$. Then, the following weak form is solved: find $v_{h}^{F}\in V^{V}+G^{V}$ and $p_{h}^{F}\in V^{P}$, such that $\forall w_{h}^{F}\in V^{V}$ and $\forall w_{h}^{C}\in V^{P}$: (23)$$\begin{array}{cc} \; & \int_{\Omega_{F}}w_{h}^{F}\rho^{F}\left({\left.\frac{\partial v_{h}^{F}}{\partial t}\right|}_{\chi }+\left(v_{h}^{F}-{\hat{v}}_{h}^{F}\right)\cdot \nabla_{x}v_{h}^{F}\right)d\Omega +\int_{\Omega_{F}}\left(\frac{1}{2}\left(\nabla_{x}w_{h}^{F}+\nabla_{x}{w_{h}^{F}}^{T}\right)\right):\sigma^{F}\left(v_{h}^{F},p_{h}^{F}\right)d\Omega \\ \; & +\int_{\Omega_{F}}w_{h}^{C}\nabla_{x}\cdot v_{h}^{F}d\Omega +\sum_{k=1}^{n_{\mathrm{el}}^{f}}\int_{\Omega_{F}^{k}}\tau_{\mathrm{SUPG}}\left(\left(v_{h}^{F}-{\hat{v}}_{h}^{F}\right)\cdot \nabla_{x}v_{h}^{F}\right)\cdot r_{M}\left(v_{h}^{F},p_{h}^{F}\right)d\Omega \\ \; & +\sum_{k=1}^{n_{\mathrm{el}}^{f}}\int_{\Omega_{F}^{k}}\tau_{\mathrm{PSPG}}\left(\frac{\nabla_{x}w_{h}^{C}}{\rho^{F}}\right)\cdot r_{M}\left(v_{h}^{F},p_{h}^{F}\right)d\Omega =0, \end{array}$$
where $w^{F}$ and $w^{C}$ denote the weight functions, $n_{\mathrm{el}}^{f}$ is the number of fluid elements, $\Omega_{F}^{k}$ denotes the *k*-th fluid element, and $\tau_{\mathrm{SUPG}}$ and $\tau_{\mathrm{PSPG}}$ are SUPG and PSPG parameters. For details on choosing them, see [34]. $r^{M}$ is defined as
(24)$$r^{M}\left(v_{h}^{F},p_{h}^{F}\right)=\rho^{F}\left({\left.\frac{\partial v_{h}^{F}}{\partial t}\right|}_{\chi }+\left(v_{h}^{F}-{\hat{v}}_{h}^{F}\right)\cdot \nabla_{x}v_{h}^{F}\right)-\nabla_{x}\cdot \sigma^{F}\left(v_{h}^{F},p_{h}^{F}\right).$$

The function space, $V^{V}$, is defined as
(25)$$V^{V}=\left\{w_{h}^{F}|\exists e^{V}={\left[e_{1x}^{V}\;e_{1y}^{V}\;\cdots \;e_{Ix}^{V}\;e_{Iy}^{V}\;\cdots \;e_{n^{F}x}^{V}\;e_{n^{F}y}^{V}\right]}^{T}\in R^{2n^{F}},w_{h}^{M}=N^{F}e^{V},e_{Ix}^{V}=e_{Iy}^{V}=0\;\mathrm{for}\;I\in \eta^{\mathrm{FSI}}\right\}.$$

$G^{V}=N^{F}g^{V}$ is a lifting function. $g^{V}$ is defined as
(26)$$g^{V}={\left[g_{1x}^{V}\;g_{1y}^{V}\;\cdots \;g_{Ix}^{V}\;g_{Iy}^{V}\;\cdots \;g_{n^{F}x}^{V}\;g_{n^{F}y}^{V}\right]}^{T},\left\{\begin{array}{cc} {\left[g_{Ix}^{V}\;g_{Iy}^{V}\right]}^{T}={\hat{v}}_{h}^{F}\left(x_{I}^{M}\right) & I\in \eta^{\mathrm{FSI}},\\ {\left[g_{Ix}^{V}\;g_{Iy}^{V}\right]}^{T}=0 & I\notin \eta^{\mathrm{FSI}}. \end{array}\right.$$

The function space, $V^{P}$, is defined as
(27)$$V^{P}=\left\{w_{h}^{C}|\exists e^{P}={\left[e_{1}^{P}\;\cdots \;e_{I}^{P}\;\cdots \;e_{n^{F}}^{P}\right]}^{T}\in R^{n^{F}},w_{h}^{C}=N^{P}e^{P}\right\}.$$

With the explicit treatment of the advection velocity and the backward Euler method, Equation (Equation 25) is discretized as follows:(28)$${\left[e^{V}\;e^{P}\right]}^{T}\left(\left[\begin{array}{cc} K^{vv} & K^{vp}\\ K^{pv} & K^{pp} \end{array}\right]\left[\begin{array}{c} d_{t+\Delta t}^{V}\\ d_{t+\Delta t}^{P} \end{array}\right]-\left[\begin{array}{c} f^{V}\\ f^{P} \end{array}\right]\right)=0\;\left(\forall e^{P}\in R^{n^{F}},\forall e^{V}\in R^{2n^{F}},e_{Ix}^{V}=e_{Iy}^{V}=0\;\mathrm{for}\;I\in \eta^{\mathrm{FSI}}\right)$$
where $K^{vv}\in R^{2n^{F}\times 2n^{F}}$, $K^{vp}\in R^{2n^{F}\times n^{F}}$, $K^{pv}\in R^{n^{F}\times 2n^{F}}$, $K^{pp}\in R^{n^{F}\times n^{F}}$, $f^{v}\in R^{2n^{F}}$, and $f^{p}\in R^{n^{F}}$ are the resulting matrices and vectors after the discretization. After the flow analysis, the fluid force on the FSI interface, denoted by $F^{\mathrm{FSI}}$, is calculated using the equilibrium of forces (Equation (Equation 9)). The flow analysis procedure, F, is performed as follows: (29)$$\begin{array}{c} {F_{h}^{\mathrm{FSI}}}_{t+\Delta t}=F\left({x_{h}^{M}}_{t+\Delta t},{\hat{v}}_{h}^{F}t+\Delta t\right). \end{array}$$

In the structural analysis, the FEM is employed for spatial discretization. To consider geometrical nonlinearity, the total Lagrange formulation is employed. The fluid force is imposed as a Neumann boundary condition in the structural analysis. In addition, the Dirichlet boundary condition (Equation (Equation 8)) is prescribed on edge DA. Then, the following weak form is solved: find $u_{h}^{S}\in V^{S}+G^{S}$, such that $\forall w_{h}^{S}\in V^{S}$: (30)$$\int_{{\Omega_{S}}_{0}}w_{h}^{S}\rho_{0}^{S}\frac{D^{2}u_{h}^{S}}{Dt^{2}}d\Omega +\int_{{\Omega_{S}}_{0}}\delta E_{h}:C:E_{h}d\Omega =\int_{{\Gamma_{\mathrm{FSI}}^{\prime }}_{0}}w_{h}^{S}F_{h}^{\mathrm{FSI}}d\Omega ,$$
where $\delta E_{h}$ is $\delta E_{h}=\frac{1}{2}\left(\nabla_{X}w_{h}^{S}+{\left(\nabla_{X}w_{h}^{S}\right)}^{T}+{\left(\nabla_{X}w_{h}^{S}\right)}^{T}\nabla_{X}u_{h}^{S}+{\left(\nabla_{X}u_{h}^{S}\right)}^{T}\nabla_{X}w_{h}^{S}\right)$ and $w^{S}$ denotes the weight function. $\Gamma_{\mathrm{FSI}}^{\prime }$ is composed of three edges AB, BC, and CD, as shown in Figure 1. The function space, $V^{S}$, is defined as
(31)$$V^{S}=\left\{w_{h}^{S}|\exists e^{S}={\left[e_{1x}^{S}\;e_{1y}^{S}\;\cdots \;e_{Ix}^{S}\;e_{Iy}^{S}\;\cdots \;e_{n^{S}x}^{S}\;e_{n^{S}y}^{S}\right]}^{T}\in R^{2n^{S}},w_{h}^{S}=N^{S}e^{S},e_{Ix}^{S}=e_{Iy}^{S}=0\;\mathrm{for}\;I\in \eta^{S}\right\},$$
where $\eta^{S}$ is a set of structural nodes that are placed on edge DA. $G^{S}=N^{S}g^{S}$ is a lifting function. $g^{S}$ is defined as
(32)$$g^{S}={\left[g_{1x}^{S}\;g_{1y}^{S}\;\cdots \;g_{Ix}^{S}\;g_{Iy}^{S}\;\cdots \;g_{n^{S}x}^{S}\;g_{n^{S}y}^{S}\right]}^{T},{\left[g_{Ix}^{S}\;g_{Iy}^{S}\right]}^{T}=U\;\mathrm{for}\;\mathrm{all}\;I.$$

By introducing the finite element discretization and the Newmark-β method and applying the Newton–Raphson method, we obtain the following incremental discrete equation: (33)$$\begin{array}{cc} \; & {e^{S}}^{T}\left({K^{S}}^{\left(i\right)}{\Delta d^{S}}^{\left(i+1\right)}-{f^{S}}^{\left(i\right)}\right)=0\;\left(\forall e^{S}\in R^{2n^{S}},e_{Ix}^{S}=e_{Iy}^{S}=0\;\mathrm{for}\;I\in \eta^{S}\right),\\ \; & \;{d_{t+\Delta t}^{S}}^{\left(i+1\right)}={\Delta d^{S}}^{\left(i+1\right)}+{d_{t+\Delta t}^{S}}^{\left(i\right)}, \end{array}$$
where the superscript *i* represents the iteration count. $K^{S}\in R^{2n^{S}\times 2n^{S}}$ and $f^{S}\in R^{2n^{S}}$ are the resulting matrix and vector after the discretization. In the present study, the initial value of $d_{t+\Delta t}^{S}$ is $d_{t+\Delta t}^{S}{}^{\left(0\right)}=d_{t}^{S}-g_{t}^{S}+g_{t+\Delta t}^{S}$. The structural analysis procedure, S, is performed as follows:(34)$$\begin{array}{c} {u_{h}^{S}}_{t+\Delta t}=S\left({F_{h}^{\mathrm{FSI}}}_{t+\Delta t}\right). \end{array}$$

From Equations (Equation 23), (Equation 31), and (Equation 36), it can be observed that the target coupled problem is equivalent to the following nonlinear equation: (35)$${u_{h}^{S}}_{t+\Delta t}=S\left(F\left(M\left({u_{h}^{S}}_{t+\Delta t}\right)\right)\right).$$

To solve Equation (Equation 37), the Broyden method, a quasi-Newton method, is employed.

## 3. POD-Galerkin Framework

In the POD-Galerkin method, low-dimensional subspaces of $V^{M}$, $V^{V}$, $V^{P}$, and $V^{S}$ are constructed. Then, $V^{M}$, $V^{V}$, $V^{P}$, and $V^{S}$ in the weak forms (Equations (Equation 19), (Equation 25), and (Equation 32)) are replaced with these subspaces. In Section 3.1, we explain how to construct the subspace. To avoid redundancy, here, we focus on the construction of the subspace of $V^{S}$. Section 3.2 presents a flowchart of the proposed POD-Galerkin FSI analysis.

Variables appearing in the dimensional reduction of $V^{M}$, $V^{V}$, $V^{P}$, and $V^{S}$ have the superscripts *M*, *V*, *P*, and *S*, respectively.

### 3.1. Snapshot POD

In this subsection, we explain how to construct a subspace of $V^{S}$ from the snapshots, denoted by $e_{\left(1\right)}^{S},e_{\left(2\right)}^{S},\cdots ,e_{\left(N_{\mathrm{snap}}^{S}\right)}^{S}$, where $e^{S}$ is defined as $e^{S}=d^{S}-g^{S}$ and the subscript (i) represents the *i*-th snapshot that is collected from the high-fidelity analysis during the offline phase. $N_{\mathrm{snap}}$ denotes the number of snapshots.

First, $e^{S}\left(t,\theta \right)$ are collected at parameter set θ and various times *t* that satisfy θ∈D and t∈T. D and T=[0,T] represent the input parameter space and the time interval of interest, respectively. Second, a set of orthonormal bases, ${\left\{\phi_{i}^{S}\right\}}_{i=1}^{k^{S}}$, is constructed such that the errors between each snapshot datum and its projection onto the subspaces spanned by ${\left\{\phi_{i}^{S}\right\}}_{i=1}^{k^{S}}$ are minimized: (36)$${\left\{\phi_{i}^{S}\right\}}_{i=1}^{k^{S}}=\underset{f_{i}\left(i=1,2,\cdots ,k^{S}\right)}{\arg\;\min}\sum_{j=1}^{N_{\mathrm{snap}}^{S}}{∥e_{\left(j\right)}^{S}-FF^{T}e_{\left(j\right)}^{S}∥}_{2}^{2}\;\mathrm{with}\;f_{i}^{T}f_{j}=\delta_{ij},\;i,j=1,\cdots ,k^{S},$$
where ${∥\cdot ∥}_{2}$ denotes the $L_{2}$ norm and $F=\left[f_{1}\;f_{2}\;\cdots \;f_{k^{S}}\right]\in R^{2n^{S}\times k^{S}}$ and *k* is the number of reduced bases. If the summation of the errors in Equation (Equation 38) is minimized sufficiently, all snapshots can be well approximated by a linear combination of the POD bases, ${\left\{\phi_{i}^{S}\right\}}_{i=1}^{k^{S}}$. Here, we make the assumption that, for any t∈T,θ∈D, $e^{S}\left(t,\theta \right)$ can be approximated via a linear combination. Under this assumption, the following subspace, ${\tilde{V}}^{S}$, can be used in Equation (Equation 32) instead of $V^{S}$.
(37)$${\tilde{V}}^{S}=\left\{w_{h}^{S}|\exists c^{S}={\left[c_{1}^{S}\;c_{2}^{S}\;\cdots \;c_{k^{S}}^{S}\right]}^{T}\in R^{k^{S}},w_{h}^{S}=N^{S}\Phi^{S}c^{S}\right\},$$
where the POD basis matrix, $\Phi^{S}$, is defined as $\Phi^{S}=\left[\phi_{1}^{S}\;\phi_{2}^{S}\;\cdots \;\phi_{k^{S}}^{S}\right]\in R^{2n^{S}\times k^{S}}$.

An effective procedure to solve the above minimization problem (Equation (Equation 38)) from the snapshot matrix $X^{S}$, which is defined as $X^{S}=\left[e_{\left(1\right)}^{S}\;e_{\left(2\right)}^{S}\;\cdots \;e_{\left(N_{\mathrm{snap}}^{S}\right)}^{S}\right]$, is through SVD: (38)$$X^{S}=V^{S}\Sigma^{S}{W^{S}}^{T},$$
where $V^{S}=\left[v_{1}^{S}\;v_{2}^{S}\;\cdots \;v_{r^{S}}^{S}\right]\in R^{2n^{S}\times r^{S}}$ contains the left singular vectors, $r^{S}$ denotes the rank of $X^{S}$, and $\Sigma^{S}=\mathrm{diag}\left(\sigma_{1}^{S},\sigma_{2}^{S},\cdots ,\sigma_{r^{S}}^{S}\right)\in R^{r^{S}\times r^{S}}$ is a diagonal matrix of singular values, in which $\sigma_{1}^{S}\geq \sigma_{2}^{S}\geq \cdots \geq \sigma_{r^{S}}^{S}>0$. $W^{S}\in R^{N_{\mathrm{snap}}^{S}\times r^{S}}$ contains the right singular vectors. A set of the first $k^{S}$ left singular vectors in $V^{S}$ associated with the $k^{S}$ largest singular values in $\Sigma^{S}$ is known to be the solution to Equation (Equation 38) (i.e., $\phi_{i}^{S}=v_{i}^{S}\;\left(i=1,\cdots ,k^{S}\right)$).

In practice, the number of POD bases, $k^{S}$, is chosen to ensure that the following reconstruction error, $\varepsilon_{\mathrm{POD}}$, for the snapshot matrix, is smaller than a given threshold: (39)$$\varepsilon_{\mathrm{POD}}=\frac{∥X^{S}-\Phi^{S}{\Phi^{S}}^{T}X^{S}{∥}_{F}^{2}}{∥X^{S}{∥}_{F}^{2}},$$
in which ${∥\cdot ∥}_{F}$ denotes the Frobenius norm.

Similar to the construction of ${\tilde{V}}^{S}$, the subspaces of $V^{V}$, $V^{P}$, and $V^{M}$, denoted by ${\tilde{V}}^{V}$, ${\tilde{V}}^{P}$, and ${\tilde{V}}^{M}$, respectively, are constructed via the snapshot POD method. High-fidelity analysis results $d^{V}-g^{V}$, $d^{P}$, and $d^{M}-g^{M}$ are collected as snapshots, and these are assembled into snapshot matrices $X^{V}\in R^{2n^{F}\times N_{\mathrm{snap}}^{V}}$, $X^{P}\in R^{n^{F}\times N_{\mathrm{snap}}^{P}}$, and $X^{M}\in R^{2n^{F}\times N_{\mathrm{snap}}^{M}}$, respectively. Then, SVD is used to determine the POD basis matrices $\Phi^{V}\in R^{2n^{F}\times k^{V}}$, $\Phi^{P}\in R^{n^{F}\times k^{P}}$, and $\Phi^{M}\in R^{2n^{F}\times k^{M}}$. ${\tilde{V}}^{V}$, ${\tilde{V}}^{P}$, and ${\tilde{V}}^{M}$ are defined as
(40)$${\tilde{V}}^{V}=\left\{w_{h}^{V}|\exists c^{V}={\left[c_{1}^{V}\;c_{2}^{V}\;\cdots \;c_{k^{V}}^{V}\right]}^{T}\in R^{k^{V}},w_{h}^{V}=N^{F}\Phi^{V}c^{V}\right\},$$
(41)$${\tilde{V}}^{P}=\left\{w_{h}^{P}|\exists c^{P}={\left[c_{1}^{P}\;c_{2}^{P}\;\cdots \;c_{k^{P}}^{P}\right]}^{T}\in R^{k^{P}},w_{h}^{P}=N^{P}\Phi^{P}c^{P}\right\},$$
(42)$${\tilde{V}}^{M}=\left\{w_{h}^{M}|\exists c^{M}={\left[c_{1}^{M}\;c_{2}^{M}\;\cdots \;c_{k^{M}}^{M}\right]}^{T}\in R^{k^{M}},w_{h}^{M}=N^{F}\Phi^{M}c^{M}\right\}.$$

### 3.2. POD-Galerkin FSI Analysis

In the POD-Galerkin FSI analysis, $V^{M}$, $V^{V}$, $V^{P}$, and $V^{S}$ in the weak forms (Equations (Equation 19), (Equation 25), and (Equation 32)) are replaced with ${\tilde{V}}^{M}$, ${\tilde{V}}^{V}$, ${\tilde{V}}^{P}$, and ${\tilde{V}}^{S}$, respectively. Hence, $d^{M}$, $d^{V}$, $d^{P}$, and $d^{S}$ are approximated in ROM as
(43)$$d^{M}\approx \Phi^{M}a^{M}+g^{M},\;d^{V}\approx \Phi^{V}a^{V}+g^{V},\;d^{P}\approx \Phi^{P}a^{P},\;d^{S}\approx \Phi^{S}a^{S}+g^{S},$$
where $a^{M}\in R^{k^{M}}$, $a^{V}\in R^{k^{V}}$, $a^{P}\in R^{k^{P}}$, and $a^{S}\in R^{K^{S}}$ are the coefficient vectors of the reduced-order approximation.

Figure 2 shows a flowchart of the proposed POD-Galerkin FSI analysis system. For the development of the proposed system, in the present study, we use our in-house code, which was verified in our previous studies [18,20]. In the high-fidelity analysis, the FSI problem is modeled as a nonlinear equation with the unknown variable of the structural displacement, as shown in Equation (Equation 37). Meanwhile, in the ROM analysis, the unknown variable is the reduced-order structural displacement, $a^{S}$. As shown in Figure 2, the dimensional reduction is applied during the fluid mesh update, fluid analysis, and structural analysis. In the reduced-order mesh update, the following equation is solved: (44)$${c^{M}}^{T}{\Phi^{M}}^{T}K^{M}\left(\Phi^{M}a^{M}+g^{M}\right)=0\;\left(\forall c^{M}\in R^{k^{M}}\right).$$

In the reduced-order fluid analysis, the following equation is solved: (45)$${\left[c^{V}\;c^{P}\right]}^{T}{\left[\begin{array}{cc} \Phi^{V} & O\\ O & \Phi^{P} \end{array}\right]}^{T}\left(\left[\begin{array}{cc} K^{vv} & K^{vp}\\ K^{pv} & K^{pp} \end{array}\right]\left[\begin{array}{c} \Phi^{V}a_{t+\Delta t}^{V}+g_{t+\Delta t}^{V}\\ \Phi^{P}a_{t+\Delta t}^{P} \end{array}\right]-\left[\begin{array}{c} f^{V}\\ f^{P} \end{array}\right]\right)=0\;\left(\forall c^{P}\in R^{k^{P}},\forall c^{V}\in R^{k^{V}}\right).$$

In the reduced-order structural analysis, the following equation is solved: (46)$$\begin{array}{cc} \; & {c^{S}}^{T}{\Phi^{S}}^{T}\left({K^{S}}^{\left(i\right)}\Phi^{S}{\Delta a^{S}}^{\left(i+1\right)}-{f^{S}}^{\left(i\right)}\right)=0\;\left(\forall c^{S}\in R^{k^{S}}\right),\\ \; & {a_{t+\Delta t}^{S}}^{\left(i+1\right)}={\Delta a^{S}}^{\left(i+1\right)}+{a_{t+\Delta t}^{S}}^{\left(i\right)},\;{d_{t+\Delta t}^{S}}^{\left(i+1\right)}=\Phi^{S}{\Delta a^{S}}^{\left(i+1\right)}+{d_{t+\Delta t}^{S}}^{\left(i\right)}. \end{array}$$

The initial value of $a_{t+\Delta t}^{S}$ is ${a_{t+\Delta t}^{S}}^{\left(0\right)}=a_{t}^{S}$. The initial value of $d_{t+\Delta t}^{S}$ is the same as that in the high-fidelity analysis: $d_{t+\Delta t}^{S}{}^{\left(0\right)}=d_{t}^{S}-g_{t}^{S}+g_{t+\Delta t}^{S}$.

## 4. Numerical Example

In this section, the 2D FSI problem of simplified flapping motion, which is described in Section 2, is considered. In the example, we investigate the effectiveness of the proposed ROM when the online prediction model parameters are not within the offline training parametric sets, as is often the case in practical product design optimization.

### 4.1. Analysis Setting

The material properties of the plate and the surrounding fluid are as follows: Poisson’s ratio is 0.3, Young’s modulus is $1.0\times 10^{9}\;\mathrm{GPa}$, the density of the plate is $7.0\times 10^{3}\;\mathrm{kg}/m^{3}$, the density of the fluid is $1.0\;\mathrm{kg}/m^{3}$, and the viscosity is $1.0\times 10^{-3}\;\mathrm{kg}/m\cdot s$. The timestep is $1.0\times 10^{-4}\;s$. The Newmark-β parameters β and γ are 0.3025 and 0.6, respectively. For the discretization of the fluid domain, 3-node triangular finite elements are used. The numbers of nodes and elements in the fluid mesh are 24,865 and 48,520, respectively. For the discretization of the structural domain, 4-node quadrilateral finite elements are used. The numbers of nodes and elements in the structural mesh are 3006 and 2500, respectively.

In our target FSI problem, flapping motion is described via two parameters: the amplitude, *A*, and the frequency, *f*. The parameter space is defined as A∈{x∣0.0015≤x≤0.0025} (unit: m) and f∈{x∣80≤x≤120} (unit: 1/s).

To obtain training data, we performed nine high-fidelity simulations, and for each high-fidelity simulation, we adopted a flapping motion based on a different parameter set inside the parametric domain, as depicted in Figure 3. To construct snapshot matrices $X^{M}$, $X^{V}$, $X^{P}$, and $X^{S}$, the high-fidelity simulation results that converged at each timestep were collected as snapshots. Each high-fidelity simulation took 500 timesteps. Therefore, each snapshot matrix had 4500 (9×500) snapshots.

Figure 4 shows the relations between the number of POD bases and the reconstruction error, $\varepsilon_{\mathrm{POD}}$, for the fluid velocity, the fluid pressure, the displacement of the fluid mesh, and the structural displacement.

We chose the POD bases so that the error is less than $1.0\times 10^{-9}$ for the structural displacement and $1.0\times 10^{-6}$ for the others. As a result, the numbers of POD bases are 159, 103, 62, and 11 for the fluid velocity, the fluid pressure, the displacement of the fluid mesh, and the structural displacement, that is, $k^{V}=159$, $k^{P}=103$, $k^{M}=62$, and $k^{S}=11$, respectively.

### 4.2. Results and Discussion

#### 4.2.1. Comparison of Accuracy

As shown in Figure 3, we consider two test cases, called “Test 1“ and “Test 2”. In Test 1, the parameter set (A,f)=(0.00175,90) was used. In Test 2, the parameter set (A,f)=(0.00225,110) was used. Here, the units m and 1/s are adopted for amplitude, *A*, and frequency, *f*, respectively. Figure 5 and Figure 6 visualize the distribution of the velocity norm around the elastic plate at different times. These figures are the results of Test 1 and Test 2 obtained from high-fidelity analysis and POD-Galerkin analysis.

Figure 7 shows the predicted time histories of the x-displacement at point B indicated in Figure 1 for Test 1 and Test 2. Figure 8 shows the predicted time histories of the lift force for Test 1 and Test 2. The lift force is calculated by integrating the y-component of the fluid force, $F^{\mathrm{FSI}}$, over the interface, $\Gamma_{\mathrm{FSI}}$. As shown in Figure 5, Figure 6, Figure 7 and Figure 8, we can see good agreement between the results from the POD-Galerkin analysis and those from the high-fidelity analysis.

The capability of the reduced-order model for the maximum lift force was investigated under various parameter sets. In total, we performed 81 reduced-order simulations, for which the following 81 parameter sets were considered:$$\begin{array}{c} \left(A,f\right)=\left(x,y|x=0.0015+(i-1)\times 0.000125,y=80+(j-1)\times 5,\;i,j=1,2,\cdots ,9\right), \end{array}$$
where the units m and 1/s are adopted for amplitude, *A*, and frequency, *f*, respectively. Figure 9a,b show the maximum lift force results from the high-fidelity analysis and ROM analysis, and Figure 9c shows the error associated with the reduced-order models when compared with the high-fidelity models. We can see from Figure 9c that the prediction results obtained from the ROM analysis agree well with the prediction via the high-fidelity analysis over all the considered parametric domains. In most cases, the maximum error is less than 10%. When f=115 and f=120, some cases show a large error. However, the amplitude is quite low in these cases. Therefore, the absolute error of these cases is small.

#### 4.2.2. Computational Time

Table 1 shows a comparison of the computational time of the high-fidelity analysis and POD-Galerkin analysis, where we have listed the total CPU time for 1000 timesteps, CPU time per Broyden iteration step for the fluid mesh update and fluid analysis, and CPU time per Newton–Raphson iteration step for the structural analysis. When calculating CPU time, we ran our computer program on a PC with 32 GB of memory and an Intel Core i9-10900K 3.70 GHz processor. In both the high-fidelity and the ROM analysis, constructing and solving matrix equations are extremely time-consuming.

Because the dimensions of the system equations in POD-Galerkin analysis are $k^{M}$, $k^{V}+k^{P}$, and $k^{S}$ (=62, 262, and 11) for the fluid mesh update, the fluid analysis, and the structural analysis, respectively, it takes much less time to solve the algebraic equations compared to solving the high-fidelity system equations with dimensions $2n^{F}$, $3n^{F}$, and $2n^{S}$ (=49,730, 74,595, and 6012). However, in the POD-Galerkin method, the following three matrix multiplication procedures are time-consuming: $$\begin{array}{c} {\Phi^{M}}^{T}K^{M}\Phi^{M},\;{\left[\begin{array}{cc} \Phi^{V} & O\\ O & \Phi^{P} \end{array}\right]}^{T}\left[\begin{array}{cc} K^{vv} & K^{vp}\\ K^{pv} & K^{pp} \end{array}\right]\left[\begin{array}{cc} \Phi^{V} & O\\ O & \Phi^{P} \end{array}\right],{\Phi^{S}}^{T}{K^{S}}^{\left(i\right)}\Phi^{S}. \end{array}$$

Among these three, the first two matrix multiplication procedures need to be computed at every Broyden iteration, and the other is computed at every Newton–Raphson iteration. As a result, for the present example, the overall computational time of the POD-Galerkin analysis is about 44% of that of the high-fidelity analysis.

## 5. Conclusions

For the prediction of FSI phenomena in flapping motion, conventional high-fidelity nonlinear numerical analysis is extremely time-consuming. To accelerate the simulation, in this work, POD-Galerkin-method-based ROM was introduced into the Dirichlet–Neumann partitioned iterative FSI analysis. In our proposed ROM method, the snapshot data of fluid velocity, fluid pressure, structural displacement, and displacement for the fluid mesh update were collected from the high-fidelity analysis. Then, the POD bases were constructed via SVD, and the discrete equation was projected onto a much smaller dimension via the Galerkin method. The matrix multiplication operations in the POD-Galerkin method were time-consuming. However, the total CPU time of the ROM analysis was much less than that of the high-fidelity analysis because ROM drastically sped up solving the matrix equations.

In the numerical example, we considered a simplified flapping motion FSI problem parametrized via the amplitude and the frequency. The DOFs of the fluid mesh update, the fluid analysis, and the structural analysis were reduced from 49,730, 74,595, and 6012 to 62, 262, and 11, respectively. We showed that with such a reduced-order model, the FSI phenomenon of flapping motion is well captured, and the maximum lift force was estimated with less than 20% error, while 44% of the high-fidelity CPU time was consumed.

In the present study, we handled the flapping motion in a non-flowing fluid, which is a simplification of the hovering flight of FWMAVs. In future work, we plan to handle 3D free-flight simulations. In this case, the Reynolds number might become higher than that in hovering flight. We need to investigate how much an increase in the advection influences the accuracy and stability of the ROM analysis.

In addition, to enhance the acceleration of the simulation due to the POD-Galerkin method, further reduction techniques must be introduced. We plan to incorporate hyper-reduction methods [8] into the reduced-order FSI analysis proposed in the present study.

The contribution of the present study is only to show the possibilities of the ALE-based reduced-order FSI analysis system. It does not include a discussion on whether the interface-tracking method or the interface-capturing method is suitable for flapping problems in the context of ROM. In future work, we will develop the reduced-order FSI analysis system based on the interface-capturing method. Then, we will compare it with the system developed in the present study.

## Figures and Tables

**Figure 1 biomimetics-08-00523-f001:**
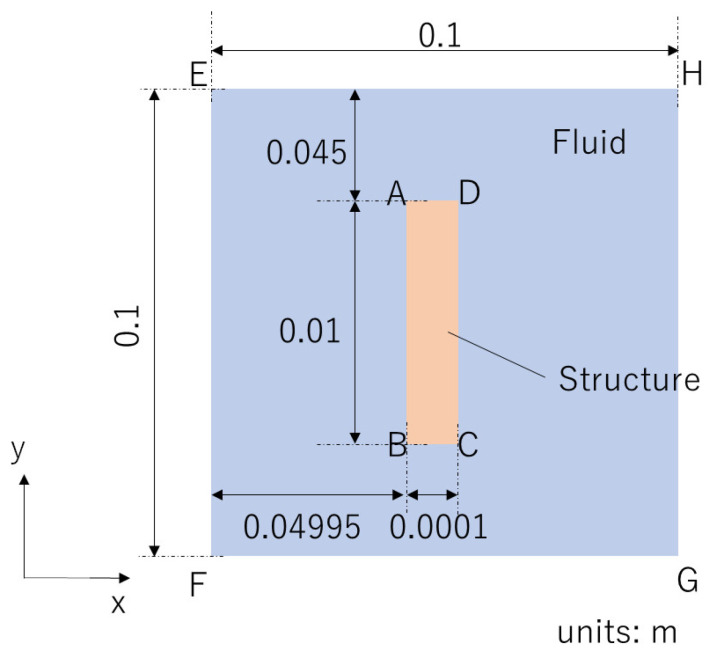
Schematic view of the target problem.

**Figure 2 biomimetics-08-00523-f002:**
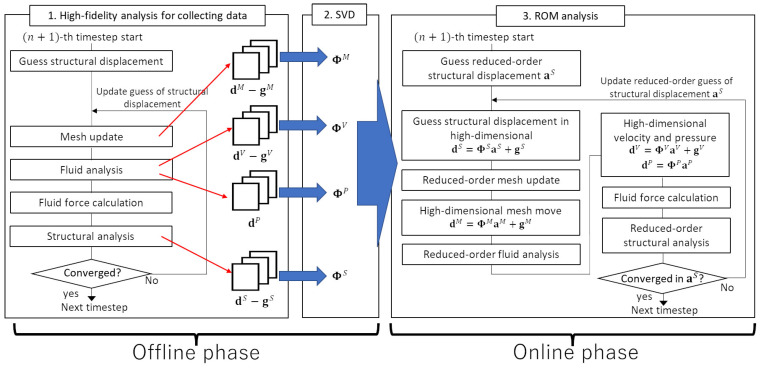
Flowchart of the proposed POD-Galerkin FSI analysis system.

**Figure 3 biomimetics-08-00523-f003:**
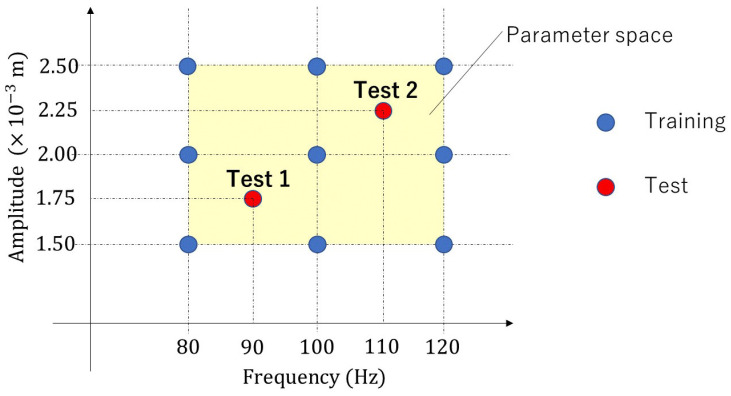
Parameter space: nine parameter sets are for offline training, and two parameter sets are for online testing.

**Figure 4 biomimetics-08-00523-f004:**
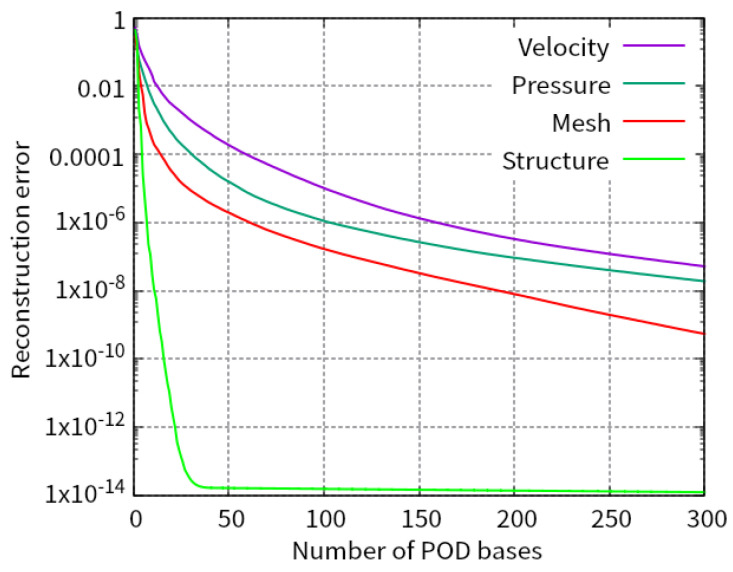
Relations between the number of POD bases and the reconstruction error.

**Figure 5 biomimetics-08-00523-f005:**
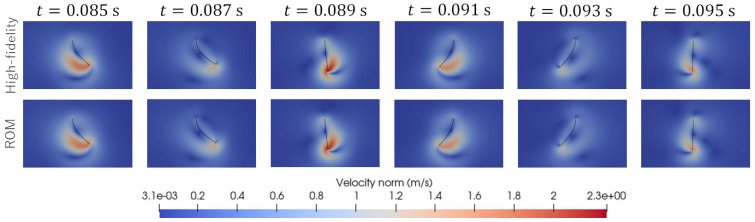
Comparison of the predicted distribution of the velocity norm at different times for Test 1.

**Figure 6 biomimetics-08-00523-f006:**
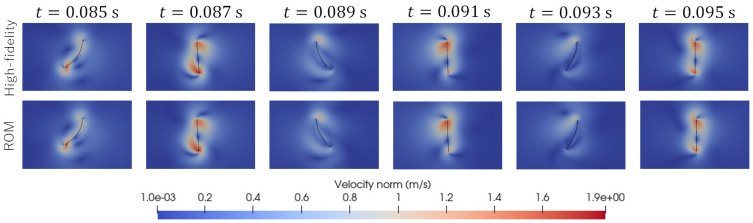
Comparison of the predicted distribution of the velocity norm at different times for Test 2.

**Figure 7 biomimetics-08-00523-f007:**
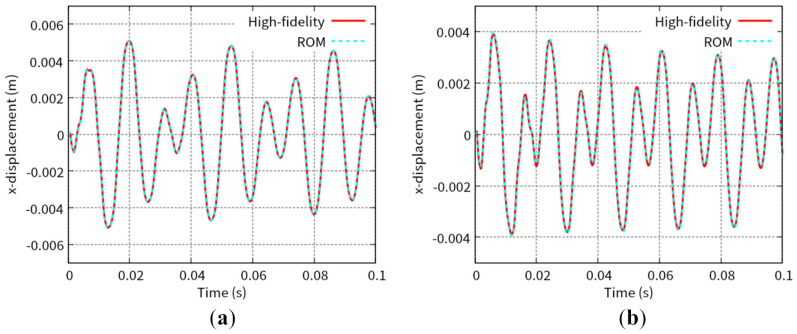
Time history of the x-displacement at point B in Figure 1 for (**a**) Test 1 and (**b**) Test 2.

**Figure 8 biomimetics-08-00523-f008:**
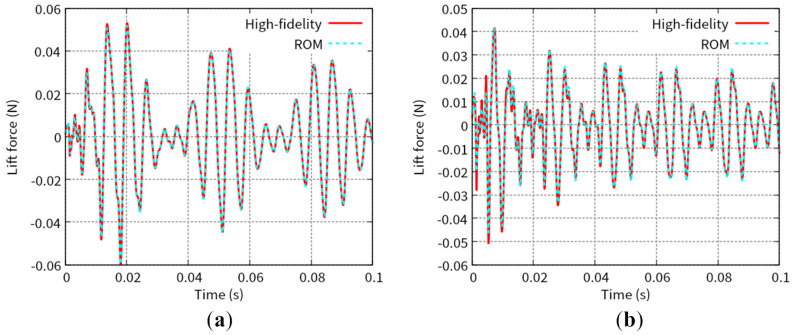
Time history of the lift force for (**a**) Test 1 and (**b**) Test 2.

**Figure 9 biomimetics-08-00523-f009:**
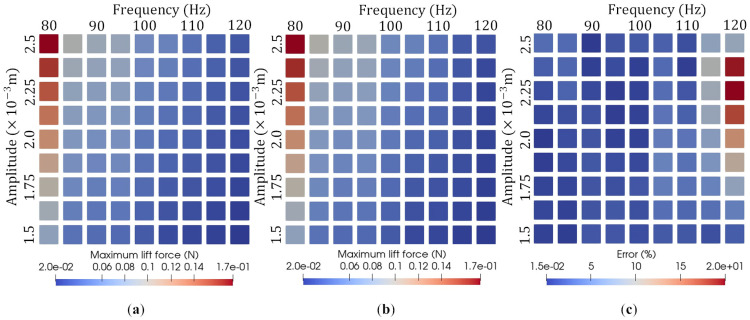
Maximum lift force prediction results for 81 different online simulation parameter sets using (**a**) high-fidelity analysis and (**b**) ROM analysis on various parameter sets, and (**c**) the error of the ROM analysis relative to the high-fidelity analysis.

**Table 1 biomimetics-08-00523-t001:** Comparison of CPU time (unit: s).

Procedure		High-Fidelity	POD-Galerkin
Fluid mesh update	Constructing matrix equation	$3.5\times 10^{-3}$	$1.7\times 10^{-2}$
	Solving matrix equation	$9.9\times 10^{-2}$	$5.3\times 10^{-5}$
Fluid analysis	Constructing matrix equation	$1.0\times 10^{-2}$	$1.1\times 10^{-1}$
	Solving matrix equation	$1.8\times 10^{-1}$	$3.4\times 10^{-4}$
Structural analysis	Constructing matrix equation	$8.5\times 10^{-4}$	$1.2\times 10^{-3}$
	Solving matrix equation	$5.5\times 10^{-3}$	$2.1\times 10^{-6}$
Total CPU time for 1000 timesteps		$9.9\times 10^{2}$	$4.4\times 10^{2}$

## Data Availability

The corresponding author may be contacted for access to the data.

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
