# Peer review of "POD-Galerkin FSI Analysis for Flapping Motion"

_biomimetics, 2023, doi:10.3390/biomimetics8070523_

Round 1
Reviewer 1 Report
Comments and Suggestions for Authors
The authors should focus the discussion of the results on the modal response of the plate. It appears to me that the plate moves according to the first mode only, thus making the whole analysis very simple. Torsional mode is absent because you chose a very high aspect ratio. The third mode?
So I would say discuss more the effect of the third mode (visible in the displacement trend) in terms of flow field effect. In few words: please add more physical description of the problem.
Comments on the Quality of English Language
Overall it's ok, maybe some minor check on some sentences that appear a bit too colloquial (see the introduction)
Reviewer 2 Report
Comments and Suggestions for Authors
Title: POD-Galerkin FSI analysis for flapping motion
[GENERAL COMMENT]
The topic of the article is very interesting, but I do not see a relevant original contribution to be published. I see an existing theoretical development in accordance with the bibliography of the topic but not a practical case with scientific solidity.
To make this statement I base on:
(i) The first contact with your target problem is Figure 1 which is not scaled (look at the ratio of the 0.01 dimension and the 0.001 dimension). This first graphical impression of your target problem has a poor initial scientific view.
(ii) I do not know with what software the numerical simulations were carried out because it was not indicated in your manuscript.
(iii) The study of meshing is key to obtaining adequate results. A real 3D geometry can be treated as 2D as you say (line 209). But I don't see a study of the influence of the mesh on the convergence in the numerical results. This point is fundamental, and I do not see it in your manuscript.
(iv) Since the results are not justified by the absence of the aforementioned meshing study, nor do I see what software was used to carry out the numerical simulations, I cannot know if the conclusions are supported by the results.
[MINOR COMMENTS/SUGGESTIONS]
[PRELIMINARY SECTION] A preliminary section of abbreviations/acronyms and units is suggested.
[line 81] Our interest. This is a very vague statement, please specify the interest(s).
[Figure 1-line 91] This Figure 1 is not scaled, at least indicate that it is not scaled in the manuscript.
[Section 2.2.1, Section 2.2.2, Section 2.2.3 and Section 2.2.4] You must indicate the units and incorporate them in the preliminary section too.
[line 171] Two ]]. Please delete one.
[Figure 3-line 273] You must use 2 decimal places to be consistent with the Y axis format (Amplitude mm).
ENGLISH STYLE
Moderate Editing of English language required.
OVERALL RECOMENDATION
Reconsider after major revision.

Comments on the Quality of English Language
Moderate Editing of English Language required
Round 2
Reviewer 2 Report
Comments and Suggestions for Authors
Title: POD-Galerkin FSI analysis for flapping motion
GENERAL COMMENT 2nd Round
Authors have justified their decisions based on my comments and, when appropriate, have incorporated them into this final version of the manuscript.
ENGLISH STYLE
Minor editing of English language required.
OVERALL RECOMMENDATION
Accept in present form.

Comments on the Quality of English Language
Minor editing of English language required.